META-RESEARCH ARTICLE

# High prevalence of articles with image-related problems in animal studies of subarachnoid hemorrhage and low rates of correction by publishers

**René Aquarius**[1]*, **Merel van de Voort**[1,2], **Hieronymus D. Boogaarts**[1],
**P. Manon Reesink**[1,2], **Kimberley E. Wever**[2]

1 Department of Neurosurgery, Radboud University Medical Center, Nijmegen, Gelderland, The Netherlands, 2 Department of Anesthesiology, Pain and Palliative Medicine, Radboud University Medical Center, Nijmegen, Gelderland, The Netherlands

* rene.aquarius@radboudumc.nl

## Abstract

Scientific progress relies on science's capacity for self-correction. If erroneous articles remain unchallenged in the publication record, they can mislead future research and undermine evidence-based decision-making. All articles included in a systematic review of animal studies on early brain injury after subarachnoid hemorrhage were analyzed for image-related issues. We included 608 articles, of which 243 articles were identified as problematic (40.0%). Of the 243 problematic articles, 55 (22.6%) have been corrected, 7 (2.9%) have received an expression of concern, 5 (2.1%) were marked with the Taylor & Francis under investigation pop-up, and 19 (7.8%) were retracted. In 9 of the 55 corrected articles (16.4%), new problems were found after correction or not all issues were resolved in the correction. Most (n = 213, 87.7%) problematic articles had a corresponding author affiliated to an institute from China. Our results show that the self-correcting mechanisms in science have stalled in this field of research. Our findings provide insight in the prevalence of image-related issues and can help publishers to take appropriate action. We can only uphold science's capacity for self-correction when problematic articles are actively identified by peers, and when publishers take swift and adequate action to repair the scientific record.

## Introduction

Scientific progress relies on its ability to be cumulative, which in turn depends on science's capacity for self-correction. If erroneous articles remain unchallenged in the publication record, they can mislead future research and undermine evidence-based decision-making. While conducting a systematic review in biomedicine, we encountered an alarmingly high rate of problematic articles, making it impossible to synthesize results across studies.

**Data availability statement:** All data are freely available in the Supporting information, and on Zenodo: https://doi.org/10.5281/zenodo.17192613.

**Funding:** The grant proposal was written by H.D.B. and was given to the Department of Neurosurgery of the Radboud University Medical Center (ZonMw, #114024137, https://www.zonmw.nl/nl). The funder required methodological guidance from a qualified meta-science expert. This was K.E.W., who is a coauthor of this manuscript. This was a mandatory step to ensure high-quality work. The funder had no further role in study design, data collection and analysis, decision to publish, or preparation of the manuscript.

**Competing interests:** We have read the journal's policy and the authors of this manuscript have the following competing interests: R.A.—PROSPERO administrator (no payment). The PROSPERO record of this review was handled by an independent administrator who is not involved in this review. Ongoing collaborator of Imagetwin and has been given free access in return. M.v.d.V.—no competing interests. H.D.B.—consultant for Stryker neurovascular. Fees are paid to the Neurosurgery department of the Radboud University Medical Center. P.M.R.—no competing interests. K.E.W.—PROSPERO administrator (no payment). The PROSPERO record of this review was handled by an independent administrator who is not involved in this review. Ongoing collaborator of Imagetwin and has been given free access in return.

**Abbreviation:** PRISMA, Preferred Reporting Items for Systematic Reviews and Meta-Analyses.

Our review originally aimed to identify promising interventions for early brain injury, a severe complication after subarachnoid hemorrhage [1,2]. Because effective interventions to reduce early brain injury are currently lacking [2,3], we set out to perform a systematic review of animal studies in this field, a strategy recommended to aid in selecting drugs with the highest likelihood of success in clinical trials.

The initial research question of our systematic review was: Can any intervention reduce early brain injury in animal models of subarachnoid hemorrhage? However, after title-abstract screening, we had identified hundreds of relevant animal experiments, in which hundreds of compounds were reported to successfully reduce early brain injury after experimental subarachnoid hemorrhage. The apparent preclinical success of this many compounds seemed implausible considering the lack of any clinically effective therapy for early brain injury [2,3], which triggered us to investigate the trustworthiness of the evidence base. Upon closer inspection, we found inappropriate image duplication and manipulation in several included studies, i.e., images were (partially) reused to represent different experimental conditions.

Given the implications of these findings, we redirected our efforts to systematically quantify the extent of image duplication and manipulation in this evidence base, employing both manual and AI-assisted image duplication detection. Our findings reveal a concerning level of erroneous images, accompanied by a failure of scientific publishing to adequately self-correct.

## Methods

This review was pre-registered in PROSPERO (ID: CRD42022347561) and is reported according to the Preferred Reporting Items for Systematic Reviews and Meta-Analyses (PRISMA) guidance [4]. This project was funded by ZonMw (project number 114024137).

### Changes to the pre-registered protocol

The alarming presence of inappropriate (image) duplication and manipulation we encountered during the screening phase, made us abandon our initial review question. This led to the following protocol amendments:

- Review question. Our initial review question ("Can any intervention reduce the severity of early brain injury in animal models of subarachnoid hemorrhage?") was changed to: "In the evidence base for the efficacy of any intervention to reduce early brain injury in animal models of subarachnoid hemorrhage, how many articles contain problematic images?"

- Screening phases. We did not perform full-text screening and continued the project with all articles included during title-abstract screening.

- Data extraction. We did not extract any of the predefined study characteristics or outcome data. Instead, we collected data regarding inappropriate image duplication and manipulation by investigating this ourselves and through PubPeer, an online platform that facilitates post-publication review of scientific literature [5].

- Data synthesis. We did not perform any of the planned early brain injury outcome data syntheses. Instead, we analyzed how many publications contained inappropriate image duplication and manipulation, which journals and publishers were affected by these problematic articles, and what the country of affiliation for each corresponding author was.

- Risk of bias assessment. We did not perform a risk of bias assessment of the articles because we did not perform any outcome data synthesis for which the risk of bias assessment could be relevant.

## Search strategy

A comprehensive search was conducted on February 10th, 2023, on Medline (via PubMed) and EMBASE (via Ovid), using thesaurus and free text terms related to "subarachnoid hemorrhage", "early brain injury", and a search filter for animal studies [6]. The full search string is presented in S1 Table. Reference lists of relevant reviews found through title-abstract screening were assessed to identify additional relevant articles.

## Study selection

All retrieved records were imported into EndNote (v20.3, Clarivate Analytics, United States) to remove duplicates and subsequently uploaded to Rayyan (Rayyan.ai, Rayyan Systems, United States) for screening. Two reviewers (M.v.d.V. and P.M.R.) independently screened all records for eligibility, based on title and abstract. Articles were included if they described the effect of an intervention on early brain injury-related outcomes within 72 hours of subarachnoid hemorrhage induction in an animal model. The following exclusion criteria were used:

1. not an original, full-length research article,

2. not an animal study,

3. no subarachnoid hemorrhage induction,

4. no outcome assessment within 72 hours (the window for early brain injury),

5. no intervention against early brain injury tested, and

6. knock-out animals only (as the proposed intervention should be translatable to humans).

Reviewers were blinded to each other's decisions. Disagreements were primarily resolved through discussion. If consensus could not be reached, a third reviewer (R.A.) acted as an arbiter.

## Study characteristics

Bibliographic data such as author(s), journal, publisher, country of affiliation of corresponding author(s), and year of publication were extracted from each included article. For the year of publication, we extracted the most recent publication date available. This was usually the date of physical publication, which typically includes the volume number. If the article was not physically published (yet), the year of publication associated with the electronic publication date was extracted.

## Image duplication and manipulation

Image duplications and manipulations are strong indicators of serious issues in the research process that can result in retraction [7]. Typical examples include reuse of western blot bands and histological images labeled as different experimental conditions. These issues suggest either deliberate manipulation or a complete failure of data oversight, both of which seriously undermine the credibility of the article.

On July 17th, 2023, R.A., K.E.W., P.M.R., and M.v.d.V. collectively performed a visual inspection of western blot images from a random sample of 80 articles (using the =RAND() function in Excel). Immediate findings of inappropriate duplication of western blots in this sample prompted R.A. to further inspect these articles. This led to the detection of several other image types with inappropriate duplication, which were confirmed by all other authors. This assessment was completed on September 17th, 2023.

Detecting image duplications by eye is extremely labor intensive, making it unfeasible to assess all articles within a reasonable timeframe. Moreover, it is difficult to detect image duplication between articles by eye. We therefore initiated a collaboration with Imagetwin (Austria, https://imagetwin.ai) [8]. This software can detect duplicated images and image elements within a figure and between figures within an article. Furthermore, it can detect duplicated images and image elements between figures of different articles, by comparing the images of the uploaded PDF to Imagetwin's proprietary database (containing ~75 million scientific images at the time of writing). We started our assessment of the included articles on February 6th, 2024, and we completed this process on October 16th, 2024. During this period, all included articles (and supplementary files, when available) were assessed multiple times, since Imagetwin's detection algorithm improved over time, and its database grew from 50 to 75 million scientific images.

We expected that some image duplications between articles in our set would not be detected by Imagetwin if the image in question was not included in their database. Imagetwin, therefore, facilitated a custom assessment comparing all included articles (and supplementary files, when available) to each other. This assessment was performed on November 11th, 2024.

## Outcome measures and reporting

To determine whether an article was suspicious, we postulated 4 yes/no questions:

1. Was the article retracted, independent of our findings?

2. Was the article corrected, independent of our findings?

3. Did we identify (a) new issue(s)?

4. Was the article flagged on PubPeer by somebody else?

If the answer to one (or more) questions was yes, we flagged the article as suspicious. All suspicious articles were then reviewed by R.A. and K.E.W. to determine if they were truly problematic, which was defined as an article having a true image-related issue, a retraction for any reason, or an article having a true non-image-related issue found by chance. Articles for which concerns were dispelled after contact with authors, journal editors or careful assessment, were not labeled as problematic. The remaining articles were labeled as problematic. For this set of articles, we calculated the following descriptive statistics:

1. Total number of problematic articles in the evidence base.

2. Occurrence of image duplication within a figure.

3. Occurrence of image duplication between figures of the same article.

4. Occurrence of image duplication between figures of different articles.

5. Occurrence of other issues.

6. Country of affiliation for corresponding author(s) for all (problematic) articles.

7. Total number of (problematic) articles per journal and publisher.

8. Number and type of editorial actions taken so far.

During this systematic review, we continuously monitored whether included articles were corrected, retracted, or flagged on PubPeer by others. The final assessment of all 608 included articles was performed from July 28th to July 30th, 2025, by R.A.

**Reporting of issues**

Any issues identified in the included articles were reported on PubPeer by R.A. (not anonymized). The first PubPeer comment was posted on August 2nd, 2023, and the last on July 30th, 2025. Any issues identified were also reported to representatives of the journal (preferably the editor-in-chief) and/or the publisher (preferably a research integrity team member).

## Results

### Study inclusion

The study selection procedure is depicted in Fig 1. Our comprehensive search yielded a total of 2068 articles, and an additional 61 were identified from reference lists of relevant reviews. A total of 1,276 unique articles underwent title and abstract screening, during which 668 articles were excluded using the predefined exclusion criteria. Thus, 608 articles, which were published between 1966 and 2024, were included in the study. The majority ($n = 565$, 92.9%) was published in the last 15 years (≥2010) (Fig 1, and https://doi.org/10.5281/zenodo.17192613).

Due to the timing of our comprehensive search (February 2023), 2022 is the last year that we assessed in full. This explains the relatively low number of articles published in 2023 included in our study. Due to the allocation of a physical volume number after being electronically available in 2023, 3 of the included articles were officially listed as having been published in 2024.

### Suspicious articles

We identified 250 articles (41.1%) as suspicious, meaning they were flagged either because (1) they were retracted, independent of our findings ($n = 3$), (2) they were corrected, independent of our findings ($n = 11$), (3) we identified (a) new issue(s) ($n = 231$), or (4) they were flagged on PubPeer by somebody else ($n = 6$). All suspicious articles were published between 2008 and 2023 (https://doi.org/10.5281/zenodo.17192613). Of these, seven articles (2.8%; article IDs 012, 045, 120, 214, 278, 286, 412) were not labeled as problematic after contact with editors, authors, or careful consideration (S2 Table).

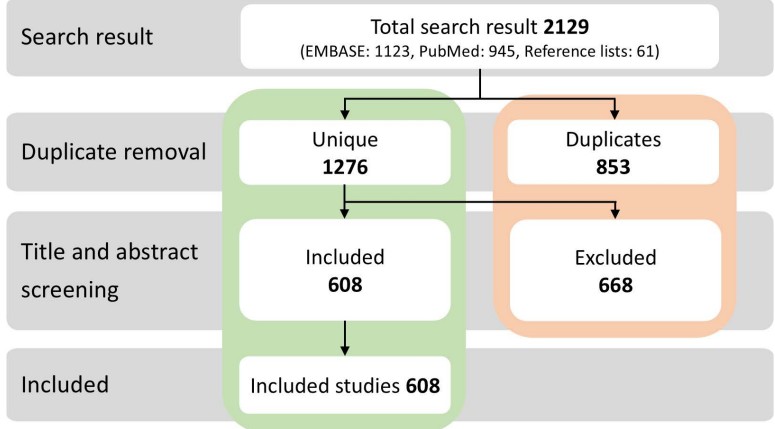

**Fig 1. Chart detailing the number of articles that were included and excluded in our study.**

## Prevalence of problematic articles

Out of 608 included articles, 243 were problematic (40.0%), which were all published between 2008 and 2023 (Fig 2, and https://doi.org/10.5281/zenodo.17192613). The number of articles published on the topic increased gradually at first (<1990–2013) and was followed by a more rapid increase between 2014 and 2022. The problematic articles showed a partially similar trend, with a gradual increase (2008–2014), followed by a rapid increase (2014–2017). However, the numbers then appear to decrease and plateau to around 25 problematic articles published every year (2018–2022).

Of the 243 problematic articles, 231 (95.1%) were identified by us, and 5 were identified by other PubPeer users (2.1%). For 7 problematic articles (2.9%), there was no accompanying PubPeer post.

Of the 243 problematic articles, 239 had image-related issues (98.4%). Note that some articles and figures contained more than one type of issue.

A total of 359 figures in 239 articles were affected in one out of 4 ways:

1. Inappropriate overlap within a single figure: 102 articles (113 figures).

2. Inappropriate overlap between figures in the same article: 39 articles (80 figures).

3. Inappropriate overlap between figures of different articles: 133 articles (181 figures).

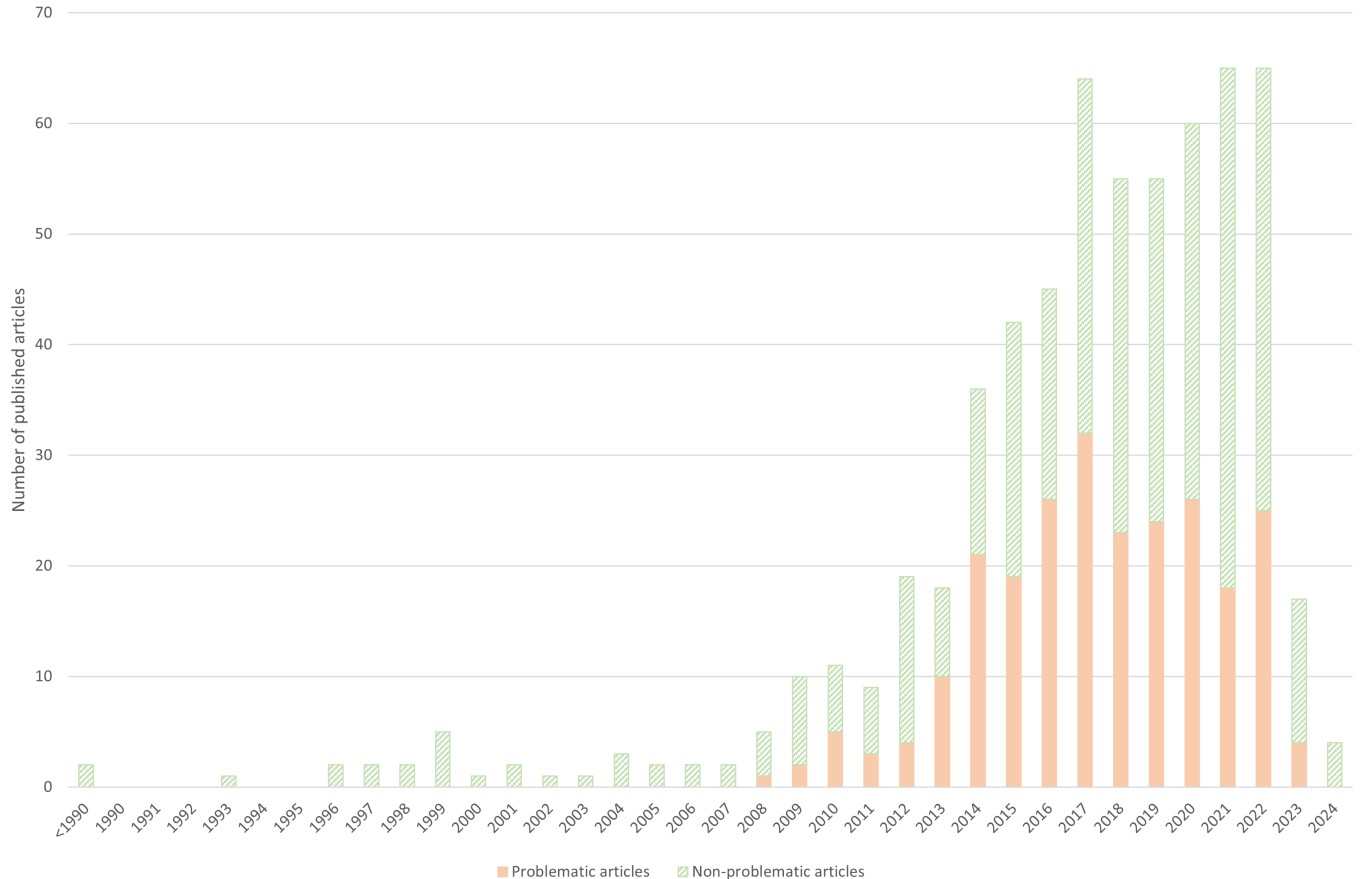

**Fig 2. Number of included articles per publication year.** Green, dashed bar segments represent non-problematic articles. Orange bar segments represent problematic articles. The data underlying this Figure can be found in https://doi.org/10.5281/zenodo.17192613.

4. Inappropriate Western blot splicing: 4 articles (4 figures).

In four articles, we or others found issues not related to images:

1. Signs that the peer-review process was compromised (2 articles).

2. Text referring to an ischemic stroke model instead of hemorrhagic stroke model (1 article).

3. Duplicated values found in tables between two articles (1 article included in our study and 1 article not included in our study).

## Corresponding authors

Of the 608 included articles, 565 articles had one (or more) corresponding author(s) with 1 country of affiliation and 43 articles had corresponding authors from 2 affiliation countries. Corresponding authors affiliated to institutes in China ($n$ = 453 articles, 74.5%), the United States of America ($n$ = 101 articles, 16.6%), and Japan (27 articles, 4.4%) were most common (S3 Table).

Of the 243 problematic articles, 213 (87.7%) had a corresponding author affiliated to an institute in China, 30 articles (12.3%) had a corresponding author affiliated to an institute in the United States of America, and 8 articles (3.3%) had a corresponding author affiliated to an institute in Taiwan (S3 Table).

## Journals and publishers affected

The five journals with the highest numbers of problematic articles were *Molecular Neurobiology* (Springer, 13 articles), *Brain Research* (Elsevier, 10 articles), *Stroke* (Lippincott Williams & Wilkins, 10 articles), *Journal of Neuroinflammation* (BioMed Central, 9 articles), and *Neuroscience Letters* (Elsevier, 9 articles). These 51 articles accounted for 21.0% of all problematic articles identified (Fig 3).

The highest percentage of problematic articles was found in the following list of journals. In six journals, 100% of the included articles were problematic: *Bioengineered* (Taylor & Francis, 3 articles), *International Immunopharmacology* (Elsevier, 2 articles), *International Journal of Medical Sciences* (Ivyspring International Publisher, 2 articles), *International Journal of Molecular Medicine* (Spandidos Pub., 2 articles), *Journal of Clinical Medicine* (MDPI, 2 articles), and *Turkish Neurosurgery* (Turkish Neurosurgical Society, 2 articles). In two journals, 83.3% of the included articles were problematic: *Acta Neurochirurgica* (Springer, 5/6 articles) and *Cell Death and Disease* (Nature Publishing Group, 5/6 articles). In one journal, 80.0% of the included articles were problematic: *FASEB Journal* (Wiley, 4/5 articles). These 27 articles accounted for 11.1% of all problematic articles identified (Fig 3).

Details on the number of problematic and non-problematic articles per publisher can be found in S1 Fig.

## Editorial actions

Journal editors and publishers have taken corrective actions for some of the problematic articles. Post-publication editorial actions were logged up to July 30th, 2025 (Table 1).

Fifty-five of 243 problematic articles (22.6%) were corrected. Forty-six corrections were a direct result of our findings and 9 corrections were not related to our findings. Of the 46 corrections that resulted due to our findings, 1 was initially performed without a correction notice (stealth correction [9]), which was rectified months later, when a correction notice was added after we notified the publisher of this omission (Table 1). One article was corrected twice. In nine corrected articles (16.4%), we either found new problems after correction or noticed that not all issues were resolved in the correction (https://doi.org/10.5281/zenodo.17192613).

Seven out of 243 problematic articles (2.9%) received an expression of concern and 5 out of 243 problematic articles (2.1%) were marked with the Taylor & Francis under investigation pop-up. All articles with an expression of concern or an under investigation pop-up received those due to our findings (Table 1).

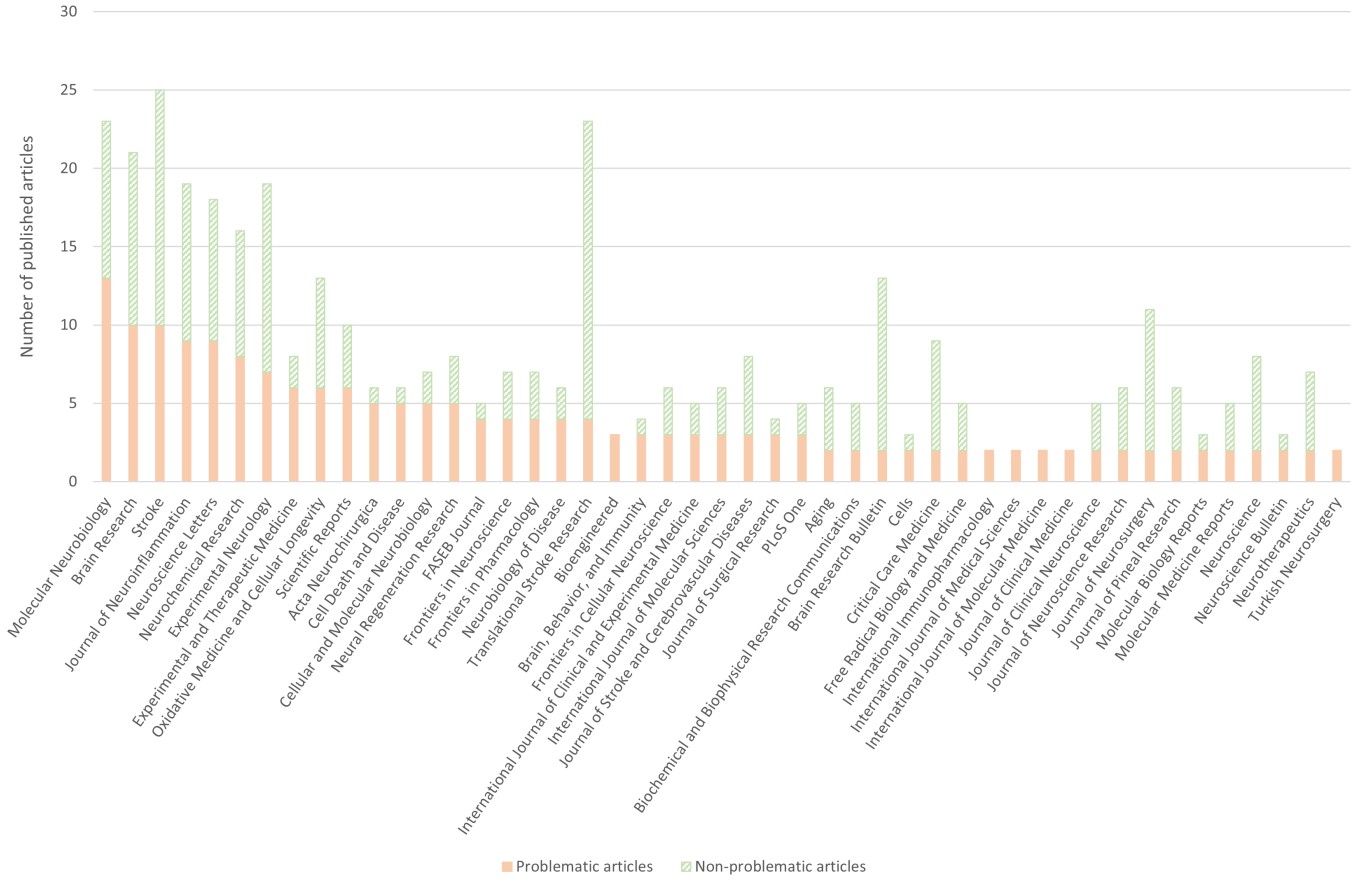

**Fig 3. Number of problematic and non-problematic articles for journals with 2 or more problematic articles.** Green, dashed bar segments represent non-problematic articles. Orange bar segments represent problematic articles. Journal names on the *X* axis are primarily ranked according to the number of problematic articles. If the number of problematic articles is the same for two or more journals, the journals are alphabetically ordered. The data underlying this Figure can be found in https://doi.org/10.5281/zenodo.17192613.

**Table 1. Overview of editorial actions up to July 30th 2025.**

| Editorial action | Editorial action caused by | Number of problematic articles |
|---|---|---|
| Correction | Correction—due to our findings | 46* |
| | Correction—not due to our findings | 9 |
| | *Subtotal* | *55* |
| Expression of concern | Expression of concern—due to our findings | 7 |
| | *Subtotal* | *7* |
| Under investigation | Under investigation pop-up—due to our findings** | 5 |
| | *Subtotal* | *5* |
| Retraction | Retraction—due to our findings | 16 |
| | Retractions—not due to our findings | 3 |
| | *Subtotal* | *19* |

The data underlying this Table can be found in https://doi.org/10.5281/zenodo.17192613.

*In one article, an image was initially replaced without a correction notice ("stealth correction"), which we reported to the publisher.

**Taylor & Francis uses an "under investigation" pop-up on their website.

Nineteen out of 243 problematic articles (7.8%) were retracted; 16 retractions occurred due to our findings, while 3 articles were retracted not due to our findings (Table 1).

### Other articles affected

In the process of this review, we also identified 36 problematic articles that were not included in our systematic review, but contained (partial) image overlap with articles that were included in our systematic review. Of these 36 articles, 10 were retracted: 6 due to our findings and 4 not due to our findings (https://doi.org/10.5281/zenodo.17192613).

## Discussion

In this systematic review of animal studies evaluating interventions to reduce the severity of early brain injury after subarachnoid hemorrhage, we demonstrate a high percentage of problematic articles in the evidence base (243/608 included articles; 40.0%).

We observed a sudden increase in the number of publications on animal studies investigating early brain injury after subarachnoid hemorrhage, which is in line with a recent bibliometric analysis of clinical and preclinical articles in this field [10]. It seems that the increase was prefaced by a few highly-cited articles expressing the need for more research on the topic [11,12]. While this call to action was meant to attract scientists with genuine interest in the topic, we hypothesize that it also provided an opportunity for bad actors to publish problematic articles. The rapid growth of the field and the novelty of the research probably made it hard to detect problematic articles for both researchers and publishers.

### Mistake or misconduct?

Inappropriate image duplication and manipulation is a problem within the scientific literature and can indicate mistakes or even fraud. While distinguishing between honest mistakes and deliberate misconduct on a case-by-case basis is challenging, the widespread nature of these issues across multiple institutions makes it unlikely that these were all innocent errors. We therefore regard all problematic articles as potential misconduct cases requiring thorough investigation. To ensure transparency, we have detailed all problematic articles with accompanying PubPeer links in a datafile available on Zenodo (https://doi.org/10.5281/zenodo.17192613).

### The tip of the iceberg

While 40% of the included articles were flagged as problematic, we have several reasons to propose that this is a conservative estimate. First, both visual and AI-driven detection methods have limitations. Imagetwin proved instrumental in identifying image duplications; however, it is not flawless. For example, some problematic articles were identified only by eye, as Imagetwin failed to flag them.

Second, presenting images from the same histological sample as different experimental groups can only be detected if there is overlap between the fields of view. When fields of view of the same histological sample are not overlapping, inappropriate image usage might be suspected, but will remain undetectable.

Last, we focused on inappropriate image duplication, yet there are several other indicators of scientific misconduct which we have not assessed. Examples include the citation of retracted articles [13], the inclusion of "sneaked references" [14], "peer-review milling" [15,16], or misidentification of scientific instruments [17]. Detection requires specialized tools, topic-specific knowledge, access to certain data and a substantial time investment.

Therefore, the number of problematic articles is likely to be even higher than the 40.0% reported in our results. We abandoned  our planned evidence synthesis as we felt unsure how useful such a synthesis would be.

### Research integrity problems—A systemic issue

Estimates of the prevalence of inappropriate image duplication in (biomedical) research remains uncertain and are dependent of the body of literature that is being investigated. Reports are sparse and cover widely different literature samples.

Out of >20,000 articles from 40 scientific journals, 4% contained problematic figures [18], while Danish researchers detected inappropriate image duplication in 19% of preclinical depression publications [19]. Image-related issues were identified in 6.1% of the assessed articles published in *Molecular and Cellular Biology* [20] and in 16% of articles published in *Toxicology Reports* [21]. Finally, in a sample of articles published in the journal *Bioengineered,* >25% contained inappropriate image duplication [22]. A synthesis of the sparse data estimates the combined misconduct rate (including fabrication, falsification, and plagiarism) to be 14%, 1 in 7 research articles [23]. The 40% prevalence observed in our study far exceeds these figures, suggesting an alarming level of integrity issues in the preclinical subarachnoid hemorrhage literature.

The prevalence of other issues, such as plagiarism and methodological issues, although variable, is in line with our findings. Plagiarism has been found in 11% to 42% of articles, depending on the investigated body of literature [24,25]. In hijacked journals, where the website of a legitimate journal has been cloned to deceive authors and databases, plagiarism can even be as high as 66% [26]. Recent examples of methodological issues include incorrect nucleotide sequences, and the use of nonverifiable and unknown cell lines, which were reported in up to 56% and 14% of the articles [27,28].

### Scientific integrity issues in China

Most (87.7%) problematic articles in our systematic review had a corresponding author affiliated to an institute in China. These findings are in line with several other results, such as a review of retracted articles in the field of neurology, in which authors affiliated to an institute in China had the highest number of retracted articles (31%) [29]. Another report pointed out that about 8,200 out of 9,600 (85%) retracted Hindawi articles has at least one coauthor with a Chinese affiliation [30].

An analysis of retracted biomedical articles from China published in 2017, revealed that serious research misconduct, such as plagiarism, faked peer-review or (suspected) fraud, was often listed as the reason for retraction [31]. As a countermeasure, cash-based publication incentives (an important driver of scientific misconduct [32]) were banned in China in 2020 [33]. However, in a survey, published in 2024 and conducted among residents at 17 tertiary hospitals in southwest China, 53.7% of respondents admitted to having committed at least one form of research misconduct [34]. Specifically, the article reported that 49.0% of the respondents admitted to "falsifying research data, materials, literature or annotations, or fabricating research results" [34].

A possible explanation for the persisting scientific integrity issues in Chinese academia could lie in the "Double First-Class University Initiative," which puts intense pressure on researchers to increase their research output in order to improve university rankings [35]. Several scholars from China, however, do not (fully) agree that the "Double First-Class University Initiative" has such far-reaching consequences and point out that improving research integrity takes time and effort [36].

### The self-correcting nature of science

The self-correcting nature of science has stalled in the preclinical subarachnoid hemorrhage field. Of the 243 problematic articles found, 231 (95.1%) had not been flagged as problematic prior to our investigation, even though some were published over a decade ago.

We have only seen post-publication, editorial action for a limited number of articles thus far. Examples of slow, opaque, and inconsistent correction of the scientific record have been described previously [37]. It is therefore of the utmost importance that publishers use our data to readily address the concerns raised in a transparent and consistent manner. Recommendations from experts might further help during this process [38].

Errors have already occurred during the process of correcting the scientific record. We have demonstrated that 9 out of 55 corrected articles (16.4%) had remaining image-related problems after the correction was issued. Publishers and editorial boards thus need to improve the process of post-publication investigation and decision-making to take the correct editorial decision. This might be hampered by editors who might have cooperated with authors of problematic articles [39], or who might be receiving bribes from paper mills [40].

We understand that investigating integrity cases can be extremely complex, and that merely reporting an issue on Pub-Peer does not warrant immediate editorial action. However, failing to act on these issues in an adequate and timely fashion perpetuates the use of potentially erroneous data, causing further damage to the research ecosystem and beyond.

Researchers will likely spend time reading and reviewing possible untrustworthy articles. For example, the review article by Lauzier and colleagues [2] cites 10 problematic articles identified by us. If these articles were corrected or retracted, the validity of their review could be undermined. Furthermore, research funding may have been used for projects based on flawed literature. The costs could be considerable: a search on the NIH "Reporter" website on June 26th, 2025, revealed that projects worth over 1 million USD have already been funded on hypotheses tied to literature on early brain injury after subarachnoid hemorrhage. These funds ultimately come from taxes.

Moreover, honest researchers may waste resources trying to reproduce or build upon problematic publications. They may even doubt the validity of their own results when findings fail to reproduce, which might lead to projects being abandoned and valid data remaining unpublished. This will push genuine scientific progress further out of reach. From an ethical point of view, there is a high risk that animals are unnecessarily sacrificed for experiments that have been designed based on false premises, which is counter to the guiding principles of replacement, reduction, and refinement (the 3Rs [41]).

All in all, these cascading effects of problematic articles may have hindered the development of effective interventions, potentially contributing to unnecessary morbidity and mortality among patients. As stated before, the apparent preclinical success of many compounds seems implausible considering the lack of any clinically effective therapy for early brain injury [2,3]. Finally, science risks further loss of public trust if its claim to be self-correcting cannot be upheld.

## Limitations

This study has three limitations. First, some image-related issues have probably been missed due to inherent limitations of the detection tools. As mentioned, this may have caused us to underestimate the prevalence of image duplication. Sole reliance on manual inspection is not necessarily more sensitive and is impractical given the vast number of articles reviewed. In our opinion, a hybrid approach such as employed here is currently the most effective strategy.

Second, Imagetwin has some caveats. Results can vary depending on the method used to analyze images. The sleuthing community has observed that scanning entire PDFs, individual figures, or high-resolution image files may yield different detection outcomes. For example, scanning a full PDF might result in no findings, while scanning a single high-resolution image results in the detection of an image duplication (or vice versa). In this study, we primarily employed full PDF scanning. Furthermore, the ability of Imagetwin to find overlapping elements within or between images is dependent of their detection algorithm, as well as the comprehensiveness of the Imagetwin image database. Both aspects are improving continuously.

Third, individual aspects of the study changed over time and were not always under our control. For example, our PubPeer comments were occasionally removed by the moderation team for reasons unknown, authors' responses were sometimes removed from PubPeer (either by the authors or by moderators), and we were often not notified of editorial decisions. Additionally, the Imagetwin algorithm kept improving over the course of the project. Finding true duplicated elements within or between images improved, false-positive findings decreased over time, and additional image types, such as flow cytometry plots, became available for assessment. Imagetwin algorithm updates were communicated to us by Imagetwin directly, through the Imagetwin website or social media channels. This necessitated multiple article assessments, which were time-consuming and significantly delayed the project.

## Conclusions

Our systematic review of animal studies evaluating interventions to reduce the severity of early brain injury after subarachnoid hemorrhage has resulted in the discovery of 243 problematic articles out of the 608 articles included (40.0%). Although these issues prevented us from performing evidence synthesis, our work emphasized the position of systematic

reviews as a tool to detect problematic articles. Our results show that the self-correcting nature of science has stalled in the field that investigates early brain injury after subarachnoid hemorrhage in animal models. Our research can help scientists to understand the widespread problems found in this field and aid publishers to take corrective actions. We can only uphold science's capacity for self-correction when problematic articles are actively identified by peers and when publishers take swift and adequate action to repair the scientific record.

## Supporting information

**S1 Table. Full search strings for EMBASE and PubMed.**
(DOCX)

**S2 Table. Overview of suspicious articles that were not labeled as problematic after contact with editors, authors, or careful consideration.** The data underlying this Table can be found in https://doi.org/10.5281/zenodo.17192613.
(DOCX)

**S3 Table. Number of articles per country of affiliation for corresponding authors.** *Forty-three articles had two countries of affiliation for corresponding authors. **Twenty-one articles had two countries of affiliation for corresponding authors. The data underlying this Table can be found in https://doi.org/10.5281/zenodo.17192613.
(DOCX)

**S1 Fig. Number of problematic articles and nonproblematic articles for each publisher that published 2 or more problematic articles.** Green, dashed bar segments represent nonproblematic articles. Orange bar segments represent problematic articles. Publisher names on the X axis are primarily ranked according to the number of problematic articles. If the number of problematic articles is the same for two or more publishers, the publishers are ordered alphabetically. The data underlying this Figure can be found in https://doi.org/10.5281/zenodo.17192613.
(DOCX)

## Acknowledgments

We thank Dorothy Bishop, Alison Avenell, and Otto Kalliokoski for their continued interest in the project and for their valuable input regarding data acquisition, data analysis, and manuscript drafting. We thank Patrick Starke and Imagetwin, without your help we could not have completed the project.

## Author contributions

**Conceptualization:** René Aquarius, Hieronymus D. Boogaarts, Kimberley E. Wever.

**Data curation:** René Aquarius, Merel van de Voort, P. Manon Reesink, Kimberley E. Wever.

**Formal analysis:** René Aquarius, Kimberley E. Wever.

**Funding acquisition:** Hieronymus D. Boogaarts.

**Investigation:** René Aquarius, Merel van de Voort, P. Manon Reesink, Kimberley E. Wever.

**Methodology:** René Aquarius, Kimberley E. Wever.

**Supervision:** René Aquarius, Kimberley E. Wever.

**Validation:** René Aquarius, Kimberley E. Wever.

**Writing – original draft:** René Aquarius, Kimberley E. Wever.

**Writing – review & editing:** René Aquarius, Merel van de Voort, Hieronymus D. Boogaarts, P. Manon Reesink, Kimberley E. Wever.

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
