## [Editor Report · Decision Letter 0]

13 May 2025

Dear Dr Aquarius,

Thank you for submitting your manuscript entitled "When science fails to self-correct" for consideration as a Meta-Research Article by PLOS Biology.

Your manuscript has now been evaluated by the PLOS Biology editorial staff, as well as by an academic editor with relevant expertise, and I'm writing to let you know that we would like to send your submission out for external peer review.

Once your full submission is complete, your paper will undergo a series of checks in preparation for peer review. After your manuscript has passed the checks it will be sent out for review. To provide the metadata for your submission, please Login to Editorial Manager (https://www.editorialmanager.com/pbiology) within two working days, i.e. by May 15 2025 11:59PM.

Kind regards,

Roli Roberts

Roland Roberts, PhD

Senior Editor

PLOS Biology

rroberts@plos.org

---

## [Decision Letter · Decision Letter 1]

23 Jun 2025

Dear Dr Aquarius,

Thank you for your patience while your manuscript "When science fails to self-correct" was peer-reviewed at PLOS Biology. It has now been evaluated by the PLOS Biology editors, an Academic Editor with relevant expertise, and by two independent reviewers.

You'll see that Reviewer #1 is overall positive, but wants you to tone down the language, especially unsupported assertions about the reasons for slow responses from editors. She also suggests mentioning other approaches, to avoid allowing identification of specific individuals, groups or institutions (as this might be legally actionable), and has a list of aspects that need to be substantially tightened up. Reviewer #2 is also positive, but mentions the need for better treatment of the existing literature, has a number of requests for clarification, and asks whether you have permission to quote the editors’ responses.

IMPORTANT: The Academic Editor strongly urges you to tone down your language and confine your interpretation to the evidence that you have to hand, saying "that kind of language runs the risk of turning people off." I also discussed the legal issues with my Editor in Chief, and on her advice we are likely to run the revised version past our legal counsel, so any adjustment of the language should be done in the current round of revision.

IMPORTANT: On a personal note, I should say that I have met many editors, from many organisations, and none of them could be described as inept, unwilling or corrupt; indeed they are mostly painfully intent on "doing the right thing." If you have evidence to the contrary, you should present it. Most editors, from my own experience, are simply under-resourced to cope with publication ethics cases on top of their day-job (which I agree is a problem). I've had the interesting experience of working at PLOS both before and after the creation of a dedicated Publication Ethics team; beforehand it was extremely challenging to move ethics cases forward, but this is now handled by the PE team, to everyone's benefit. That said, many cases still involve extensive institutional investigation and legal wrangling, and can take years to resolve... You may find it helpful to interview some professional editors to ask them what *they* perceive to be the barriers to the speedy resolution of publication ethics cases.

In light of the reviews, which you will find at the end of this email, we would like to invite you to revise the work to thoroughly address the reviewers' reports.

Given the extent of revision needed, we cannot make a decision about publication until we have seen the revised manuscript and your response to the reviewers' comments. Your revised manuscript is likely to be sent for further evaluation by all or a subset of the reviewers.

**IMPORTANT - SUBMITTING YOUR REVISION**

*Re-submission Checklist*

*Published Peer Review*

*PLOS Data Policy*

*Blot and Gel Data Policy*

Sincerely,

Roli Roberts

Roland Roberts, PhD

Senior Editor

PLOS Biology

rroberts@plos.org

REVIEWERS' COMMENTS:

Reviewer #1:

[identifies herself as Jennifer Byrne]

This manuscript describes a significant body of work that was undertaken to examine the integrity of literature on the use of animal models to study preclinical subarachnoid hemorrhage. I should declare that I was aware of this study prior to reviewing the manuscript, for example by inviting the first and senior authors to give a presentation of their preliminary results to a small online group. However, I had not seen this manuscript prior to being invited to review, and I did not know that it had been submitted. While being broadly familiar with the study, I have nonetheless highlighted all issues that I believe might be unclear to the diverse readership that this manuscript is likely to attract.

The manuscript could also be improved to more clearly describe the full body work that was undertaken, including describing problematic articles to editors and publishers. Some suggestions reflect the required manuscript format, where results are described before the methods. This requires presenting descriptions of basic methods and specifying some other information such as search dates in the results, as the reader won't have encountered this information previously.

In some sections, the language should be toned down or changed to be more objective.

Major issues:

Page 6: This section begins to describe corrective actions by journals and publishers, but it's currently unclear whether corrective actions were in response to the authors' descriptions of their concerns, and/or something else. Text in the methods and results described in the discussion should be moved to the results to make this clear.

The article includes some expressions and statements that should be toned down or removed: "an overwhelming presence" (page 9, page 15)- this implies that most articles were problematic; "we can confidently say that most journal editors and editors in chief appear to be either inept or unwilling" , "editors are often unaware"; "paper mill bribery may unfortunately be at play here". Please see comments below about potential liability issues.

Page 9: "We demonstrate their (ie systematic reviews) unique ability to detect problematic articles." (repeated page 15: "unique position"). In fact, at least one other approach identified the same proportions of problematic articles within the human cancer research literature. Our team found 38% of all articles published in the journal Molecular Cancer in 2020 to have wrongly identified gene sequence reagents, and 40% of all miRNA and circRNA articles published in Oncogene in 2020 to have the same problem (Pathmendra P et al. 2024). These proportions are very similar to the 40% of problematic articles identified here, showing that other approaches, ie screening every (or every relevant) article in single journals, can generate very similar/ the same results. I therefore suggest removing the word "unique". Much as I'm reluctant to suggest adding citations to our team's work, the similarity between these two sets of results seems worth considering.

Page 9: "across multiple institutions", also "cluster within certain research groups" (page 10)- this data should either be described in the Results or omitted. There could be defamation risks in identifying individual institutions or groups eg. "question the reliability of any research produced by this group" (page 10). It would be easy to identify this group from the articles flagged. It may be safer to discuss papers from a broader perspective, eg. according to country of origin. Perhaps the journal can advise here.

Page 18: The literature search was described as having been conducted in February 2023. How did this study then describe problematic articles published in 2024? (Figure 2) The low number of papers identified in 2023 should similarly be discussed, presumably in relation to the screening date.

Finally, I suggest that some consideration to the ethical consequences of problematic papers in terms of research animal use.

Minor issues:

Page 1: The current manuscript title is generic- suggest mentioning literature on early brain injury or preclinical subarachnoid hemorrhage.

Page 2: "All articles include in a systematic review"… in the abstract, without further context, this suggests that a systematic review was conducted, when this was abandoned. Suggest rewording to make this clear.

Page 3: "the lack of any clinically effective therapy for early brain injury"- please cite references here.

Page 4: "Our comprehensive search"- please provide dates when this was conducted.

Page 4: "The previously defined exclusion criteria"- please describe- readers have not yet read the methods.

Page 4 "unflagged"- suggest replacing with "removed from the problematic article group"

Figure 2: Please add details of what's shown on the Y axis.

Figure 2: Some interesting data are shown but not described or discussed, eg increases in total numbers of papers and problematic papers over time.

Page 5: "extremely similar"- please define

Page 5: "In four articles, we or others"- please indicate which issues were found by the author team, and which were found by others (presumably highlighted on PubPeer?)

Figure 3: The legend should describe how the data on the X axis were ranked (according to number of problematic articles?). Please also add details of what's shown on the Y axis.

Page 6: "how publishers have been affected"- suggest rewording, unclear.

Page 7: "almost always"- what proportion exactly?

Page 7: "We decided to log them as an expression of concern"- this is incorrect. Published "expressions of concern" are denoted as such- the article is simply "under investigation". Articles under investigation currently represent a new category of flagged article and it's unclear whether or how these notices resolve over time, ie. an expression of concern may or may not be published as a result.

Page 9: No references were cited.

Page 9: "this makes it impossible"- in theory, a systematic review was still possible, with the problematic articles excluded. If this wasn't done, it would be helpful to explain why.

Page 10: Please cite references to support: limitations of image integrity detection methods (line 1), "repeatedly raised by Elisabeth Bik" (line 7).

Page 11: "slow, ineffective and inconsistent": the relevant data were not described in the Results.

Page 13: "while communication has significantly improved"- please provide details, since when, how?

Page 14: "the scientific record has been severely compromised"- in what field?

Page 14: "Researchers have likely spent time and resources"- providing citations of problematic papers would be a simple way to strengthen this claim.

Page 14: "different detection outcomes"- please outline what these are, otherwise the limitation of mostly using pdf scanning is unclear.

Page 14: "individual aspects of the study changed over time"- it may be worth adding something like "and were not always under our control".

Page 14: "the ImageTwin algorithm kept improving" how was this ascertained? Over what time period was this noted?

Page 15: "multiple re-analyses"- is this described in the Methods?

Page 16: "title-abstract screening"- if only titles and abstracts were screened, problematic images won't be found. Please clarify.

Page 17: "reference lists of relevant reviews" please identify/ cite the relevant literature reviews.

Page 17: what constituted an "animal study"- presumably mice and rats? Please specify. This won't be obvious to many readers.

Page 18: "knock-out animals only"- does this mean that articles were excluded if studies only examined knock-out models, and no wild-type models? Again, this won't be obvious to many readers.

Reviewer #2:

Manuscript has incredible results and it should be published. The methods are sound. Without any doubt, this manuscript highlights the challenges of paper mills and fraud in science, problems that are significantly underestimated in the scientific community.

Some comments to authors.

1.      „Screening and study inclusion" belong to Methods and Data Selection rather than Results.

2.      A literature review is lacking. Where can this manuscript be placed within the existing literature?

3.      It should be explained what an inappropriate overlap is.

4.      It is unclear what the difference is between "The five journals with the highest incidence of problematic articles were" and "The highest prevalence of problematic articles was found in the following list of journals":

5.      What is a problematic article? One that contains image duplication, or are other types of problems included as well?

6.      It is not clear what "other articles" means. From which sample?

"Other articles affected 3 In the process of this review, we also identified 37 problematic articles outside of our included 4 article set."

7.      Regarding citations from editors' answers. Was notification or permission from the editors obtained to use the quotes in the publication?

Was ethical approval received to send concerns to the journals?

This is a suggestion to evaluate the use of quotes from editors' responses, as editors may contact journals to request corrections or retractions of the present article. You may contact Lonni Besancon who has a relevant experience regarding this problem.

8.      The manuscript mentions that misconduct could be a reason for the lack of innovation in the field of treatment for hemorrhagic stroke. It would be valuable if the manuscript included more discussion of this issue, as the results are truly shocking.

---

## [Decision Letter · Decision Letter 2]

23 Sep 2025

Dear Rene,

Thank you for your patience while we considered your revised manuscript "When science fails to self-correct - Problematic articles in a systematic review of animal studies on subarachnoid hemorrhage" for publication as a Meta-Research Article at PLOS Biology. This revised version of your manuscript has been evaluated by the PLOS Biology editors, the Academic Editor and the original reviewers.

Based on the reviews, we are likely to accept this manuscript for publication, provided you satisfactorily address the remaining points raised by the reviewers and the following data and other policy-related requests:

IMPORTANT - please attend to the following:

a) Please change your Title to remove punctuation and make it more appealing. We suggest: "High frequencies of articles with image-related problems in animal studies of brain injury and low rates of correction by publishers"

b) Please attend to the remaining requests from the reviewers.

c) Please re-name your supplementary files “Supporting information file 1,” “Supporting information file 3” and “Supporting information file 4” as “Table S1,” Table S2” and “Table S3.” Rename “Supporting information file 2” as “Data S1” and rename “Supporting information file 5” as “Figure S1.”

d) Please cite the location of the data clearly in all relevant main and supplementary Figure legends, e.g. “The data underlying this Figure can be found in S1 Data” or “The data underlying this Figure can be found in https://zenodo.org/records/XXXXXXXX

e) Please make any custom code available, either as a supplementary file or as part of your data deposition.

We expect to receive your revised manuscript within two weeks.

*Published Peer Review History*

*Press*

Sincerely,

Roli

Roland Roberts, PhD

Senior Editor

rroberts@plos.org

PLOS Biology

CODE POLICY

DATA NOT SHOWN?

REVIEWERS' COMMENTS:

Reviewer #1:

[identifies herself as Jennifer Byrne]

I would like to thank the authors for their patience and attention to my many comments on the original submitted version. The revised version is a revelation, in part due to the reordering of the methods and results, which has made the manuscript far more accessible, as well as their extensive rewriting of the text.

I only have minor suggestions:

Page 6, line 21: "the first 80 articles"- it seems that 80 articles were examined (ie "a random sample of 80 articles"- is there a need to discuss "the first"?

Page 8, line 4: "were reviewed by …. to determine if they were truly problematic" -it would help if a clear definition of problematic articles could be given here, ie "which were defined as….".

Page 20, lines 2-3: "might be hampered by editors who co-operated…. receiving bribes from paper mills". While I recognise that the language has been toned down considerably, this text seemed relatively strong, particularly as this refers to a small number of incomplete corrections (9 articles, page 19). We have also described incorrect corrections (Byrne et al. 2021, Scientometrics), where we proposed that paper mills might try to manipulate post-publication correction processes. However, this could easily happen without co-operation of the editor. The incorrect corrections that we described probably reflected lack of attention to detail and/or lack of understanding of the principles of molecular biology, that probably applied to both editors and authors. In summary, while the present text could be retained, the language could be softened, ie "might have co-operated….. and/or might be receiving". Otherwise it could imply that the editors of these 9 articles may have been colluding with authors AND receiving bribes, when it's likely that neither took place.

Page 20, line 10: please remove "which"- not needed.

Reviewer #2:

Authors did significant revision of the manuscript. I would recommend to accept it with minor revisions

Some comments:

1. -"The study selection procedure is depicted in Fig. 1. Our comprehensive search yielded a total of 2068 references, and an additional 61 were identified from reference lists of relevant reviews. A total of 1276 unique references underwent title and abstract screening, during which 668 articles were excluded using the predefined exclusion criteria."

As I understand 2068 papers have been retrieved from EMBASE, etc. I wouldn't call them references because they are papers. References are what is cited in the paper. I suggest to correct and call them papers (articles) to avoid confusion. If they are indeed references, it should be additionally mentioned in methods.

2. -"A total of 359 figures were affected in one out of 4 ways:"

I would precise A total of 359 figures in 239 papers (correct if I'm wrong) were affected in one out of 4 ways:

3. "First, both visual and AI-driven detection methods have limitations. ImageTwin proved instrumental in identifying image duplications, however, it is not flawless. For example, some problematic articles were identified by eye, as ImageTwin failed to flag any of them."

As for limitations, it is important to mention that detection depends on the libraries available (if it is the case in the study. Did you screen against the whole library of ImageTwin?).

---

## [Editor Report · Decision Letter 3]

25 Sep 2025

Dear Rene,

Thank you for the submission of your revised Meta-Research Article "High prevalence of articles with image-related problems in animal studies of subarachnoid hemorrhage and low rates of correction by publishers" for publication in PLOS Biology. On behalf of my colleagues and the Academic Editor, Marcus Munafo, I'm pleased to say that we can in principle accept your manuscript for publication, provided you address any remaining formatting and reporting issues. These will be detailed in an email you should receive within 2-3 business days from our colleagues in the journal operations team; no action is required from you until then. Please note that we will not be able to formally accept your manuscript and schedule it for publication until you have completed any requested changes.

Sincerely, 

Roli

Senior Editor

PLOS Biology

rroberts@plos.org